

# Optimizing road safety: integrated analysis of motorized vehicle using lattice ordered complex linear diophantine fuzzy soft set

K. Ashma Banu[1], J. Vimala[1], Nasreen Kausar[2] and Željko Stević[3]

[1] Department of Mathematics, Alagappa University, Tamil Nadu, India
[2] Department of Mathematics, Faculty of Arts and Science, Yildiz Technical University, Istanbul, Turkey
[3] Department of Mobile Machinery and Railway Transport, Vilnius Gediminas Technical University, Vilnius, Lithuania

Corresponding authors
Nasreen Kausar,
kausar.nasreen57@gmail.com
Željko Stević,
zeljko.stevic@vilniustech.lt

## ABSTRACT

In this manuscript, we delve into the realm of lattice ordered complex linear diophantine fuzzy soft set, which constitutes an invaluable extension to the existing Fuzzy set theories. Within this exploration, we investigate basic operations such as $\oplus$ and $\otimes$, together with their properties and theorems. This manuscript is more amenable in two ways, *i.e.*, it enables real-life problems involving parametrization tool and applications with an existing order between the components of the parameter set based on the preference in the complex frame of reference. Adaptive cruise control (ACC) is a system designed for maintaining distance between two vehicles and to sustain a manually provided input speed. The purpose of cars with ACC is to avoid a collision that frequently happens nowadays, thereby improving road safety regulations amidst rising collision rates. The fundamental aim of this manuscript is to prefer an applicable car with ACC together with its latest model by defining a peculiar postulation of lattice ordered complex linear diophantine fuzzy soft set (LOCLDFSS). Emphasizing real-life applicability, we illustrate the effectiveness and validity of our suggested methodology in tackling current automotive safety concerns, providing useful guidance on reducing challenges related to contemporary driving conditions.

# INTRODUCTION

## Literature review

Proactive approaches for improving road safety are held in high regard these days. Road accidents frequently occur due to irrelevant speed, misjudging a curve, and distraction which results in a massive amount of deaths and severe injuries according to the report released by the *Ministry of Road Transport & Highways Government of India (2022)*. The analysis states that "hit from back" with 21.2 percent contributed to the largest share in total accidents and the number of accidents during 2021 by "head-on collision" constitute 18.5 percent which happens on roads with sharp curves, narrow lanes, and busy stretches. A "rear-end collision" or "hit from back" takes place when a vehicle collides with the one in

front of it due to a loss of control by the driver. Another major type of collision which is the root cause for accidents is "hit and run" (13.9 percent) referred from *Ministry of Road Transport and Highways: Government of India (2024)*. Road safety is important when driving, and it is covered in *Adedeji & Feikie (2021)*, *Kodepogu, Manjeti & Siriki, 2023*, *Al-Hussein et al. (2021)*

Advances in technology have built automobiles safer over modern times. Car accidents have been minimized because of new technology improvements like airbags and lane departure warning systems. Advanced Driver Assistant Systems (ADAS) are primarily focused on collision avoidance technologies. Investigating the variables affecting the acquisition and learning experience of cars with ADAS is discussed in *Nandavar et al. (2023)*. Some of the benefits of ADAS technology are addressed by *Masello et al. (2022)*. ACC is one feature found in cars nowadays that uses ADAS technology to help maintain a safe following distance. Like standard cruise control, adaptive cruise control (ACC) also uses sensors and cameras to detect a nearby vehicle and respond accordingly by adjusting its speed automatically. This system then helps to maintain the set input speed for safer driving. ACC is designed mainly for safety which is thoroughly explained in *Pampel et al. (2020)*. The development of ADAS technology has drawn the attention of numerous researchers (*De-Las-Heras, Sanchez-Soriano & Puertas, 2021*; *Masello et al., 2023*; *Weber et al., 2023*; *Antony & Whenish, 2021*; *Jayapal, Muvva & Desanamukula, 2023*). *Katari et al. (2024)* investigates the crucial significance of electronic control units (ECUs) and inertial measurement units (IMUs) in Advanced Driver Assistance Systems (ADAS). It specifically examines their functions, uses, regulatory frameworks, new technologies, and real-life examples. *Ali et al. (2024)* examines a simplified version of the ACC function by utilising model predictive control. *Dai & Koutsoukos (2020)* introduces a safety analysis technique for automotive control systems by employing multi-modal port-Hamiltonian systems. The method utilises passivity to demonstrate that trajectories are unable to overcome the energy barrier between safe and unsafe states.

MCDM technique is a process of finding a leading alternative amidst various aspirants in a decision. However, most of the decisions in real-life problems have substantially vague constraints. Thereby, *Zadeh (1965)* introduced the postulation of fuzzy set in which the particulars relative to Membership Class ($\widehat{MC}$) are taken in a frame of membership function. After that many of the experimenters are engaged in their extensions. *Atanassov (1986)* initiated the notion of Intuitionistic Fuzzy set ($\widehat{IFS}$) which is one of the profitable developments that is characterized by $\widehat{MC}$ and Non-Membership Class ($\widehat{NMC}$) with regard to the criteria that sum of $\widehat{MC}$ and $\widehat{NMC}$ is intrinsic to [0,1]. Nevertheless, there may be many situations in which the decision maker may provide the $\widehat{MC}$ and $\widehat{NMC}$ in such a manner that their sum is greater than 1. For an illustration, suppose a decision maker intimate his preference about the $\widehat{MC}$ as 0.7 and $\widehat{NMC}$ as 0.5. Then, it is seen that 0.7 + 0.5 exceeds 1. To prevail over this situation, *Yager (2013)* gave a notion of Pythagorean Fuzzy set ($PFS$) with regard to the criteria that the sum of the square of $\widehat{MC}$ and $\widehat{NMC}$ is intrinsic to [0,1]. As an extension, *Riaz & Hashmi (2019)* developed the conception of Linear Diophantine Fuzzy set ($\widehat{LDFS}$) by introducing reference parameters

where the decision maker has the freedom to choose $\widehat{MC}$ and $\widehat{NMC}$ without any restriction.

*Molodtsov (1999)* gave the conceptualization of Soft set $(\widehat{SS})$ with many operations defined in detail to deal with the situations involving parametrization tool for modeling uncertainty and vagueness. *Maji, Biswas & Roy (2001)* originated the postulation of Intuitionistic Fuzzy Soft set $(\widehat{IFSS})$ as a hybrid structure of $\widehat{IFS}$ and $\widehat{SS}$. He also developed this concept by defining properties based on decision-making. Many researchers have worked on the integration of $\widehat{SS}$ with fuzzy extension theory (*Jeevitha et al., 2023*; *Jafar, Muniba & Saqlain, 2023*; *Riaz & Farid, 2023*; *Vimala et al., 2023*; *Moslem et al., 2024*).

Many of the real-life problems involving 2-D information can be accessible only if the codomain of $\widehat{FS}$ is a complex number when substituted for [0,1]. As a result, *Ramot et al. (2002)* described the theory of Complex Fuzzy set $(\widehat{CFS})$ which is specified by the Complex-Valued Membership Class $(\widehat{CVMC})$ (*i.e.*), $\Gamma_X(l_p)e^{i2\pi w_{\Gamma_X}(l_p)}$ where $\Gamma_X(l_p)$ is called the amplitude term and $w_{\Gamma_X}(l_p)$ is called the phase term with regard to the criteria that $\Gamma_X(l_p), w_{\Gamma_X}(l_p) \in [0,1]$. As a prolongation, *Kamacı (2022)* established the conceptualization of Complex Linear Diophantine Fuzzy set $(\widehat{CLDFS})$ with the cosine similarity measure characterized by $\widehat{CVMC}$, Complex-Valued Non-Membership Class $(\widehat{CVNMC})$, Complex-Valued Reference Parameter corresponding to $\widehat{MC}$ and Complex-Valued Reference Parameter corresponding to $\widehat{NMC}$.

Lattice theory is one important concept for mathematicians and researchers working on uncertainties. *Mahmood et al. (2018)* habitualized the notion of Lattice Ordered Intuitionistic Fuzzy Soft set $(\widehat{LOIFSS})$ which is very useful in a specific form of decision-making problem when there exists some hierarchy between the elements of a parameter set. Many scientists have employed decision making techniques in their work to address many real-life problems (*Arockia Reeta & Vimala, 2016*; *Anusuya Ilamathi, Vimala & Davvaz, 2019*).

*Çağman & Enginoğlu (2010)* established the concept of Soft Matrix $(\widehat{SM})$ for the easy computation with the operations of $\widehat{SS}$ and are profitably used in decision-making process. As a development of $\widehat{SM}$, *Yang & Ji (2011)* instigated the idea of Fuzzy Soft Matrix $(\widehat{FSM})$. Additionally, Intuitionistic Fuzzy Soft Matrix $(\widehat{IFSM})$ was initiated by *Dhar (2016)*. The notion of Pythagorean Fuzzy Soft Matrix $(\widehat{PFSM})$ was identified by *Guleria & Bajaj (2019)* together with some operations defined. *Rajareega & Vimala (2021)* developed the idea of Complex Intuitionistic Fuzzy Soft Matrix $(\widehat{CIFSM})$. Some of the experimenters have worked on the extension of $\widehat{SM}$ and also discussed decision making techniques using Score Matrix (*Guleria & Bajaj, 2019*) and Utility Matrix (*Guleria & Bajaj, 2019*).

Commencing with the significance of road accidents, it is important to recognize their enormous influence on public safety and welfare. Every year, traffic accidents result in injuries, fatalities, and enormous financial losses, posing a serious threat to human life. It becomes essential to incorporate safety-enhancing technology like ACC in response to this urgent problem. By encouraging safer driving habits, ACC not only addresses the underlying causes of accidents but also denotes a proactive approach to accident avoidance. Therefore, the need to reduce traffic accidents emphasizes the strategic

integration of ACC technology into car selection procedures. Numerous research works utilizing fuzzy extensions seek to improve the efficiency and safety of automotive systems by modifying their parameters in response to real-time data.

*Yanmaz et al. (2020)* proposed the Interval-valued Pythagorean fuzzy edas method for effectively selecting the most suitable car to reduce accidents. The risk prioritization in self-driving cars is provided by *Karasan et al. (2020)* using the concept of $\widehat{PFS}$. *Hussain et al. (2023)* focused on the application of this approach in assessing electric cars in a 2D frame of reference. Following that, none addresses the issue of reducing car accidents in the $\widehat{LOCLDFSS}$ environment.

## Research motivation

The modeling of the MCDM problems requires a profound significance on the attributes. In particular, ordering between the parameters demands greater focus to deal with certain real-life problems. A notion called $\widehat{LOCLDFSS}$ deals with attributes (with an order existing between them), while $\widehat{CLDFS}$ does not deal with attributes. So, we have selected to study the field of $\widehat{SS}$ theory for dealing with the attributes. To minimize the road accidents that are frequently happening in India, manufacturers have become more obsessed with enhancing vehicle safety and thereby reducing injuries and fatalities. It is advantageous to make use of a car with ACC. The fundamental aim of this manuscript is to enhance the concept of $\widehat{LOCLDFSS}$ by defining some of the basic operations like $\oplus$ and $\otimes$ and an algorithm is presented together with the numerical example to exemplify the function of suggested method in the context of $\widehat{LOCLDFSS}$. As a result, more studies to prefer an applicable car with ACC by using the MCDM technique are needed to precisely capture the uncertainties. The proposed algorithm rule is made in accordance with the Score function to address issues with decision making.

## Novelty of the proposed methodology

Despite the abundance of papers on car accidents in the literature, none address the issue in a $\widehat{LOCLDFSS}$ setting. The following is a summary of some of the proposed study's innovative contributions:

1) The main advantage of this theory is to quantify uncertainty in terms of $\widehat{CVMC}$, $\widehat{CVNMC}$, and reference parameters. In addition to that the suggested method $\widehat{LOCLDFSS}$ opens up possibilities for the parametrization tool together with the order existing in them. This aspect of the suggested study eclipses earlier fuzzy sets existing in the literature.

2) To the best of our knowledge, there hasn't been any prior writing on addressing an issue of car accidents in a $\widehat{LOCLDFSS}$ setting. Because this work is the first of its type, our suggested model is a pathfinder and an exemplar in this field.

3) $\widehat{MCDM}$ problems are pervasive in human civilization and have several applications in real-world industries. We propose an $\widehat{MCDM}$ algorithm with Score Matrix under $\widehat{LOCLDFSS}$ to prefer an applicable car with ACC together with its latest model.

4) The suggested method allows decision makers to get the optimum outcomes through the use of the $\widehat{MCDM}$ algorithm.

5) Better results are obtained with the simpler, more computationally straightforward model.

**Objectives and structure of the proposed manuscript**

In light of these positive effects mentioned above, the primary objectives of this manuscript are as follows.

1) A notion of $\widehat{LOCLDFSS}$ is interpreted with few of the basic operations and associated theorems.

2) An algorithm is constructed in accordance with the Score function to address certain MCDM problems.

3) An associated case study is also provided to prefer an applicable car with ACC together with its latest model

4) A comparative analysis with the existing algorithm is discussed to scrutinize the viability of the proposed manuscript.

The structure of the manuscript is mentioned as follows: "Preliminaries" confers some of the pre-requisite definitions such as $\widehat{SS}$, $\widehat{CFS}$, $\widehat{IFS}$, $\widehat{PFS}$, $\widehat{LDFS}$, $\widehat{CLDFS}$, $\widehat{IFSS}$ and $\widehat{LOIFSS}$. "Lattice Ordered Complex Linear Diophantine Fuzzy Soft Set" grants a notion of Lattice Ordered Complex Linear Diophantine Fuzzy Soft set with few of its operations defined and associated theorems. In "$\widehat{LOCLDFSS}$-Decision Making Process" we have introduced the conception of Score Matrix and Utility Matrix for $\widehat{LOCLDFSS}$. In "MCDM Technique Based on Score Matrix and Utility Matrix" the incorporation of the MCDM technique on $\widehat{LOCLDFSS}$ to select a leading alternative amidst various aspirants in a decision was interpreted. In "Comparative Assessment with the Existing Methodology" we have discussed the comparison of a proposed algorithm method with the other different existing algorithm methods to scrutinize the viability of the proposed method.

## PRELIMINARIES

The preliminary section includes the foundational literacy like $\widehat{SS}$, $\widehat{CFS}$, $\widehat{IFS}$, $\widehat{PFS}$, $\widehat{LDFS}$, $\widehat{CLDFS}$, $\widehat{IFSS}$ and $\widehat{LOIFSS}$ that will assist to manifest the structure of the following article. on the whole, $\hat{L}$ signifies the universal set and $\hat{H}$ symbolize the parameter set.

**Definition 2.1.** *Molodtsov (1999) The Soft set Z is interpreted on $\hat{L}$ as*

$$Z = \left\{ (h, S(h))/h \in \hat{H}, S(h) \in P(\hat{L}) \right\}$$

*where $P(\hat{L})$ is known as the power set of $\hat{L}$*

**Definition 2.2.** *Ramot et al. (2002) The Complex Fuzzy set Y defined on $\hat{L}$ as*

$$Y = \left\{ \left( l_p, \left\langle \Gamma_Y(l_p) e^{i\Delta_{\Gamma_Y}(l_p)} \right\rangle \right) : l_p \in \hat{L} \right\}$$

where $\Gamma_Y(l_p)e^{i\Delta_{\Gamma_Y}(l_p)}$ (for $\Gamma_Y(l_p) \in [0,1], \Delta_{\Gamma_Y}(l_p) \in [0, 2\pi]$) denotes the $\widehat{CVMC}$ of $l_p \in \hat{L}$. The Complex Fuzzy set can be reconsidered as

$$Y = \left\{ \left( l_p, \left\langle \Gamma_Y(l_p)e^{i2\pi w_{\Gamma_Y}(l_p)} \right\rangle \right) : l_p \in \hat{L} \right\}$$

where $\Gamma_Y(l_p)e^{i2\pi w_{\Gamma_Y}(l_p)}$ (for $\Gamma_Y(l_p)$, $w_{\Gamma_Y}(l_p) \in [0,1]$) represents the $\widehat{CVMC}$ of $l_p \in \hat{L}$.

**Definition 2.3.** *Atanassov (1986) The Intuitionistic Fuzzy set X characterized on $\hat{L}$ as*

$$X = \left\{ \left( l_p, \left\langle \Gamma_X(l_p), \Delta_X(l_p) \right\rangle \right) : l_p \in \hat{L} \right\}$$

$\Gamma_X(l_p), \Delta_X(l_p) \in [0,1]$ *encompass the $\widehat{MC}$, $\widehat{NMC}$ of $l_p \in \hat{L}$ respectively with the constrains* $0 \leq \Gamma_X(l_p) + \Delta_X(l_p) \leq 1$.

**Definition 2.4.** *Yager (2013) The Pythagorean Fuzzy set D particularized on $\hat{L}$ as*

$$D = \left\{ \left( l_p, \left\langle \Gamma_D(l_p), \Delta_D(l_p) \right\rangle \right) : l_p \in \hat{L} \right\}$$

$\Gamma_D(l_p), \Delta_D(l_p) \in [0,1]$ *encompass the $\widehat{MC}$, $\widehat{NMC}$ of $l_p \in \hat{L}$ respectively with the constrains* $0 \leq (\Gamma_D(l_p))^2 + (\Delta_D(l_p))^2 \leq 1$. *The pythagorean Fuzzy Number ($\widehat{PFN}$) is given by*

$$D = \langle \Gamma_D, \Delta_D \rangle$$

**Definition 2.5.** *Riaz & Hashmi (2019) The Linear Diophantine Fuzzy set W interpreted on $\hat{L}$ as*

$$W = \left\{ \left( l_p, \left\langle \Gamma_W(l_p), \Delta_W(l_p) \right\rangle, \left\langle \alpha_W^p, \beta_W^p \right\rangle \right) : l_p \in \hat{L} \right\}$$

where $\Gamma_W(l_p), \Delta_W(l_p), \alpha_W^p$ and $\beta_W^p \in [0,1]$ encompass the $\widehat{MC}$, $\widehat{NMC}$ and reference parameters of $l_p \in \hat{L}$ respectively with the constraints $0 \leq \alpha_W^p + \beta_W^p \leq 1$ and $0 \leq \alpha_W^p \Gamma_W(l_p) + \beta_W^p \Delta_W(l_p) \leq 1$.

**Definition 2.6.** *Kamacı (2022) The Complex Linear Diophantine Fuzzy set R is defined on $\hat{L}$ as*

$$R = \left\{ \left( l_p, \left\langle \Gamma_R(l_p)e^{i2\pi(w_{\Gamma_R}(l_p))}, \Delta_R(l_p)e^{i2\pi(w_{\Delta_R}(l_p))} \right\rangle, \left\langle \alpha_R^p e^{i2\pi(w_{\alpha_R^p})}, \beta_R^p e^{i2\pi(w_{\beta_R^p})} \right\rangle \right) : l_p \in \hat{L} \right\}$$

where $\Gamma_R(l_p)e^{i2\pi(w_{\Gamma_R}(l_p))}, \Delta_R(l_p)e^{i2\pi(w_{\Delta_R}(l_p))}, \alpha_R^p e^{i2\pi(w_{\alpha_R^p})}$ and $\beta_R^p e^{i2\pi(w_{\beta_R^p})}$ denotes the $\widehat{CVMC}$, $\widehat{CVNMC}$ and complex-valued reference parameters of $l_p \in \hat{L}$ correspondingly with the constraints $0 \leq \alpha_R^p + \beta_R^p \leq 1$, $0 \leq \alpha_R^p \Gamma_R(l_p) + \beta_R^p \Delta_R(l_p) \leq 1$ and $0 \leq w_{\alpha_R^p} + w_{\beta_R^p} \leq 1$, $0 \leq w_{\alpha_R^p} w_{\Gamma_R}(l_p) + w_{\beta_R^p} w_{\Delta_R}(l_p) \leq 1$. The Complex Linear Diophantine Fuzzy Number ($\widehat{CLDFN}$) is given by

$$R = \left( \left\langle (\Gamma_R, w_{\Gamma_R}), (\Delta_R, w_{\Delta_R}) \right\rangle, \left\langle (\alpha_R, w_{\alpha_R}), (\beta_R, w_{\beta_R}) \right\rangle \right)$$

**Definition 2.7.** *Maji, Biswas & Roy (2001) The Intuitionistic Fuzzy Soft set interpreted on $\hat{L}$ by the well-set of ordered pairs as*

$$\langle A, \hat{H} \rangle = \left\{ \langle h, A(h) \rangle / h \in \hat{H}, A(h) \in \widehat{IFS}(\hat{L}) \right\}$$

$$(i.e.), A(h) = \left\{ \left( l_p, \langle \Gamma_{A(h)}(l_p), \Delta_{A(h)}(l_p) \rangle \right): l_p \in \hat{L} \right\}.$$

*where $A : \hat{H} \to \widehat{IFS}(\hat{L})$ such that $A(h) = \phi$ if $h \notin \hat{H}$ and $\widehat{IFS}(\hat{L})$ denote the collection of all intuitionistic fuzzy subsets of $\hat{L}$.*

**Definition 2.8.** *Mahmood et al. (2018) A pair $(A, \hat{H})$ called Intuitionistic Fuzzy Soft set is said to be a Lattice Ordered Intuitionistic Fuzzy Soft set over $\hat{L}$ if for $h_1, h_2 \in \hat{H}$ such that $h_1 \leq h_2 \Rightarrow A(h_1) \subseteq A(h_2)$*

$$(i.e.), \Gamma_{A(h_1)}(l_p) \leq \Gamma_{A(h_2)}(l_p)$$
$$\Delta_{A(h_1)}(l_p) \geq \Delta_{A(h_2)}(l_p), \forall \, l_p \in \hat{L}$$

**Definition 2.9.** *Zulqarnain et al. (2021) The Score function for a $\widehat{PFN}$ $D = \langle \Gamma_D, \Delta_D \rangle$ can be interpreted as*

$$\hat{S}(D) = (\Gamma_D)^2 - (\Delta_D)^2, \text{where } \hat{S}(D) \in [-1, 1].$$

**Definition 2.10.** *Kamacı (2022) The Score function for a $\widehat{CLDFN}$ $R = (\langle (\Gamma_R, w_{\Gamma_R}), (\Delta_R, w_{\Delta_R}) \rangle, \langle (\alpha_R, w_{\alpha_R}), (\beta_R, w_{\beta_R}) \rangle)$ can be particularized as*

$$\hat{S}(R) = \frac{1}{4}[(\Gamma_R - \Delta_R) + (w_{\Gamma_R} - w_{\Delta_R}) + (\alpha_R - \beta_R) + (w_{\alpha_R} - w_{\beta_R})], \quad where \quad \hat{S}(R) \in [-1, 1].$$

**Definition 2.11.** *Jayakumar et al. (2023) Let $\hat{L} = \{l_1, l_2, \ldots, l_n\}$ denotes the universal set and $\widehat{CLDFSU}(\hat{L})$ be the collection of all Complex Linear Diophantine Fuzzy subsets of $\hat{L}$. Consider a mapping $A: \hat{H} \to \widehat{CLDFSU}(\hat{L})$. Then the Complex Linear Diophantine Fuzzy Soft set $(\widehat{CLDFSS})$ determined by the well-set of ordered pairs is interpreted as*

$$\langle A, \hat{H} \rangle = \left\{ \langle h, A(h) \rangle / h \in \hat{J}, A(h) \in CLDFSU(\hat{L}) \right\}$$

$$(i.e.), A(h) = \left\{ \left( l_p, \left\langle \Gamma_{A(h)}(l_p) e^{i2\pi \left( w_{\Gamma_{A(h)}}(l_p) \right)}, \Delta_{A(h)}(l_p) e^{i2\pi \left( w_{\Delta_{A(h)}}(l_p) \right)} \right\rangle, \right.\right.$$
$$\left.\left. \left\langle \alpha^p_{A(h)} e^{i2\pi \left( w_{\alpha^p_{A(h)}} \right)}, \beta^p_{A(h)} e^{i2\pi \left( w_{\beta^p_{A(h)}} \right)} \right\rangle \right): l_p \in \hat{L} \right\}.$$

*such that $A(h) = \phi$ if $h \notin \hat{H}$. The Complex Linear Diophantine Fuzzy Soft set can also be characterized as*

$$(i.e.), A(h) = \left\{ \left( l_p, \left\langle \left( \Gamma_{A(h)}(l_p), w_{\Gamma_{A(h)}}(l_p) \right), \left( \Delta_{A(h)}(l_p), w_{\Delta_{A(h)}}(l_p) \right) \right\rangle, \right.\right.$$
$$\left.\left. \left\langle \left( \alpha^p_{A(h)}, w_{\alpha^p_{A(h)}} \right), \left( \beta^p_{A(h)}, w_{\beta^p_{A(h)}} \right) \right\rangle \right): l_p \in \hat{L} \right\}.$$

# LATTICE ORDERED COMPLEX LINEAR DIOPHANTINE FUZZY SOFT SET

In this component, a specific postulation called Lattice Ordered Complex Linear Diophantine Fuzzy Soft set was presented together with the basic operational laws such as $\oplus$ and $\otimes$ which provide more comfort potential in dealing with the limitations of values in decision-making problems.

**Definition 3.1.** *A pair $(A, \hat{H})$ called Complex Linear Diophantine Fuzzy Soft set is said to be a Lattice Ordered Complex Linear Diophantine Fuzzy Soft set $(LO\widehat{CLD}FSS)$ over $\hat{L}$ if for $h_1$, $h_2 \in \hat{H}$ such that $h_1 \leq h_2 \Rightarrow A(h_1) \subseteq A(h_2)$*

$$(i.e.), \Gamma_{A(h_1)}(l_p) \leq \Gamma_{A(h_2)}(l_p)$$
$$w_{\Gamma_{A(h_1)}}(l_p) \leq w_{\Gamma_{A(h_2)}}(l_p)$$
$$\Delta_{A(h_1)}(l_p) \geq \Delta_{A(h_2)}(l_p)$$
$$w_{\Delta_{A(h_1)}}(l_p) \geq w_{\Delta_{A(h_2)}}(l_p)$$
$$\alpha^p_{A(h_1)} \leq \alpha^p_{A(h_2)}$$
$$w_{\alpha^p_{A(h_1)}} \leq w_{\alpha^p_{A(h_2)}}$$
$$\beta^p_{A(h_1)} \geq \beta^p_{A(h_2)}$$
$$w_{\beta^p_{A(h_1)}} \geq w_{\beta^p_{A(h_2)}}, \forall l_p \in \hat{L}$$

**Definition 3.2.** *Let us consider two $LO\widehat{CLD}FSS$ $(A, \hat{H})$ and $(B, \hat{H})$. The elementary operational law $\oplus$, also known as Algebraic sum is characterized as*

$$(Z, \hat{H}) = (A, \hat{H}) \oplus (B, \hat{H})$$
$$(i.e.), Z(h) = \{(l_p, \langle(\Gamma_{Z(h)}(l_p), w_{\Gamma_{Z(h)}}(l_p)), (\Delta_{Z(h)}(l_p), w_{\Delta_{Z(h)}}(l_p))\rangle,$$
$$\langle(\alpha^p_{Z(h)}, w_{\alpha^p_{Z(h)}}), (\beta^p_{Z(h)}, w_{\beta^p_{Z(h)}})\rangle): l_p \in \hat{L}\}.$$

*where*

$$\left\{ \begin{array}{l} \Gamma_{Z(h)}(l_p) = \Gamma_{A(h)}(l_p) + \Gamma_{B(h)}(l_p) - \Gamma_{A(h)}(l_p)\Gamma_{B(h)}(l_p) \\ w_{\Gamma_{Z(h)}}(l_p) = w_{\Gamma_{A(h)}}(l_p) + w_{\Gamma_{B(h)}}(l_p) - w_{\Gamma_{A(h)}}(l_p)w_{\Gamma_{B(h)}}(l_p) \\ \Delta_{Z(h)}(l_p) = \Delta_{A(h)}(l_p)\Delta_{B(h)}(l_p) \\ w_{\Delta_{Z(h)}}(l_p) = w_{\Delta_{A(h)}}(l_p)w_{\Delta_{B(h)}}(l_p) \\ \alpha^p_{Z(h)} = \alpha^p_{A(h)} + \alpha^p_{B(h)} - \alpha^p_{A(h)}\alpha^p_{B(h)} \\ w_{\alpha^p_{Z(h)}} = w_{\alpha^p_{A(h)}} + w_{\alpha^p_{B(h)}} - w_{\alpha^p_{A(h)}}w_{\alpha^p_{B(h)}} \\ \beta^p_{Z(h)} = \beta^p_{A(h)}\beta^p_{B(h)} \\ w_{\beta^p_{Z(h)}} = w_{\beta^p_{A(h)}}w_{\beta^p_{B(h)}} \end{array} \right\}_{\{p=1,2,\ldots,n\}}$$

**Definition 3.3.** *The operational law $\otimes$, also known as Algebraic product is defined by considering two $LO\widehat{CLD}FSS$ $(A, \hat{H})$ and $(B, \hat{H})$ as*

$$(Y, \hat{H}) = (A, \hat{H}) \otimes (B, \hat{H})$$

$$(i.e.), Y(h) = \{(l_p, \langle (\Gamma_{Y(h)}(l_p), w_{\Gamma_{Y(h)}}(l_p)), (\Delta_{Y(h)}(l_p), w_{\Delta_{Y(h)}}(l_p)) \rangle,$$

$$\langle (\alpha_{Y(h)}^p, w_{\alpha_{Y(h)}^p}), (\beta_{Y(h)}^p, w_{\beta_{Y(h)}^p}) \rangle) \colon l_p \in \hat{L}\}.$$

*where*

$$\begin{cases} \Gamma_{Y(h)}(l_p) = \Gamma_{A(h)}(l_p)\Gamma_{B(h)}(l_p) \\ w_{\Gamma_{Y(h)}}(l_p) = w_{\Gamma_{A(h)}}(l_p)w_{\Gamma_{B(h)}}(l_p) \\ \Delta_{Y(h)}(l_p) = \Delta_{A(h)}(l_p) + \Delta_{B(h)}(l_p) - \Delta_{A(h)}(l_p)\Delta_{B(h)}(l_p) \\ w_{\Delta_{Y(h)}}(l_p) = w_{\Delta_{A(h)}}(l_p) + w_{\Delta_{B(h)}}(l_p) - w_{\Delta_{A(h)}}(l_p)w_{\Delta_{B(h)}}(l_p) \\ \alpha_{Y(h)}^p = \alpha_{A(h)}^p \alpha_{B(h)}^p \\ w_{\alpha_{Y(h)}^p} = w_{\alpha_{A(h)}^p} w_{\alpha_{B(h)}^p} \\ \beta_{Y(h)}^p = \beta_{A(h)}^p + \beta_{B(h)}^p - \beta_{A(h)}^p \beta_{B(h)}^p \\ w_{\beta_{Y(h)}^p} = w_{\beta_{A(h)}^p} + w_{\beta_{B(h)}^p} - w_{\beta_{A(h)}^p} w_{\beta_{B(h)}^p} \end{cases} \{p=1,2,\dots,n\}$$

**Definition 3.4.** *Let $\lambda > 0$ and $\lambda$ be real. Considering a $LOC\widehat{LD}FSS$ $(A, \hat{H})$, we define $\lambda(A, \hat{H})$*

$$(X, \hat{H}) = \lambda(A, \hat{H})$$

$$(i.e.), X(h) = \{(l_p, \langle (\Gamma_{X(h)}(l_p), w_{\Gamma_{X(h)}}(l_p)), (\Delta_{X(h)}(l_p), w_{\Delta_{X(h)}}(l_p)) \rangle,$$

$$\langle (\alpha_{X(h)}^p, w_{\alpha_{X(h)}^p}), (\beta_{X(h)}^p, w_{\beta_{X(h)}^p}) \rangle) \colon l_p \in \hat{L}\}.$$

*where*

$$\begin{cases} \Gamma_{X(h)}(l_p) = 1 - (1 - \Gamma_{A(h)}(l_p))^\lambda \\ w_{\Gamma_{X(h)}}(l_p) = 1 - (1 - w_{\Gamma_{A(h)}}(l_p))^\lambda \\ \Delta_{X(h)}(l_p) = (\Delta_{A(h)}(l_p))^\lambda \\ w_{\Delta_{X(h)}}(l_p) = (w_{\Delta_{A(h)}}(l_p))^\lambda \\ \alpha_{X(h)}^p = 1 - \left(1 - \alpha_{A(h)}^p\right)^\lambda \\ w_{\alpha_{X(h)}^p} = 1 - \left(1 - w_{\alpha_{A(h)}^p}\right)^\lambda \\ \beta_{X(h)}^p = \left(\beta_{A(h)}^p\right)^\lambda \\ w_{\beta_{X(h)}^p} = \left(w_{\beta_{A(h)}^p}\right)^\lambda \end{cases} \{p=1,2,\dots,n\}$$

**Definition 3.5.** *Let $\lambda > 0$ and $\lambda$ be real. Considering a $LOC\widehat{LD}FSS$ $(A, \hat{H})$, we define $(A, \hat{H})^\lambda$*

$$(V, \hat{H}) = (A, \hat{H})^\lambda$$

$$(i.e.),\ V(h) = \left\{ \left( l_p, \left\langle \left( \Gamma_{V(h)}(l_p), w_{\Gamma_{V(h)}}(l_p) \right), \left( \Delta_{V(h)}(l_p), w_{\Delta_{V(h)}}(l_p) \right) \right\rangle, \right.\right.$$
$$\left.\left. \left\langle \left( \alpha_{V(h)}^p, w_{\alpha_{V(h)}^p} \right), \left( \beta_{V(h)}^p, w_{\beta_{V(h)}^p} \right) \right\rangle \right): l_p \in \hat{L} \right\}.$$

*where*

$$\begin{cases} \Gamma_{V(h)}(l_p) = (\Gamma_{A(h)}(l_p))^\lambda \\ w_{\Gamma_{V(h)}}(l_p) = (w_{\Gamma_{A(h)}}(l_p))^\lambda \\ \Delta_{V(h)}(l_p) = 1 - (1 - \Delta_{A(h)}(l_p))^\lambda \\ w_{\Delta_{V(h)}}(l_p) = 1 - (1 - w_{\Delta_{A(h)}}(l_p))^\lambda \\ \alpha_{V(h)}^p = (\alpha_{A(h)}^p)^\lambda \\ w_{\alpha_{V(h)}^p} = (w_{\alpha_{A(h)}^p})^\lambda \\ \beta_{V(h)}^p = 1 - (1 - \beta_{A(h)}^p)^\lambda \\ w_{\beta_{V(h)}^p} = 1 - \left(1 - w_{\beta_{A(h)}^p}\right)^\lambda \end{cases}_{\{p=1,2,\dots,n\}}$$

**Example 3.6.** *Consider two $\widehat{LOCLDFSS}$ $(A, \hat{H})$ and $(B, \hat{H})$. The hierarchy between the parameters is $h_1 \leq h_2$. Let*

$$(A, \hat{H}) = \{A(h_1) = \{l_1, \langle (0.6, 0.5), (0.3, 0.3) \rangle, \langle (0.7, 0.6), (0.2, 0.3) \rangle$$
$$l_2, \langle (0.8, 0.7), (0.2, 0.2) \rangle, \langle (0.8, 0.8), (0.2, 0.1) \rangle\}$$
$$A(h_2) = \{l_1, \langle (0.7, 0.6), (0.1, 0.3) \rangle, \langle (0.8, 0.6), (0.2, 0.1) \rangle$$
$$l_2, \langle (0.8, 0.8), (0.2, 0.1) \rangle, \langle (0.9, 0.9), (0.1, 0.1) \rangle\}$$

*and*

$$(B, \hat{H}) = \{B(h_1) = \{l_1, \langle (0.6, 0.6), (0.3, 0.2) \rangle, \langle (0.6, 0.7), (0.2, 0.3) \rangle$$
$$l_2, \langle (0.6, 0.5), (0.2, 0.4) \rangle, \langle (0.7, 0.6), (0.2, 0.3) \rangle\}$$
$$B(h_2) = \{l_1, \langle (0.9, 0.8), (0.2, 0.2) \rangle, \langle (0.8, 0.8), (0.2, 0.1) \rangle$$
$$l_2, \langle (0.7, 0.8), (0.2, 0.1) \rangle, \langle (0.8, 0.6), (0.2, 0.3) \rangle\}$$

*and also $\lambda = 4$. Thus, we have*

1. $(A, \hat{H}) \oplus (B, \hat{H}) = \{h_1, \{l_1, \langle (0.84, 0.8), (0.09, 0.06) \rangle, \langle (0.88, 0.88), (0.04, 0.09) \rangle$
   $l_2, \langle (0.92, 0.85), (0.04, 0.08) \rangle, \langle (0.94, 0.92), (0.04, 0.03) \rangle\}$
   $\{h_2 = \{l_1, \langle (0.97, 0.92), (0.02, 0.06) \rangle, \langle (0.96, 0.92), (0.04, 0.01) \rangle$
   $l_2, \langle (0.94, 0.96), (0.04, 0.01) \rangle, \langle (0.98, 0.96), (0.02, 0.03) \rangle\}\}$
2. $(A, \hat{H}) \otimes (B, \hat{H}) = \{h_1, \{l_1, \langle (0.36, 0.30), (0.51, 0.44) \rangle, \langle (0.42, 0.42), (0.36, 0.51) \rangle$
   $l_2, \langle (0.48, 0.35), (0.36, 0.52) \rangle, \langle (0.56, 0.48), (0.36, 0.37) \rangle\}$
   $\{h_2 = \{l_1, \langle (0.63, 0.48), (0.28, 0.44) \rangle, \langle (0.64, 0.48), (0.36, 0.19) \rangle$
   $l_2, \langle (0.56, 0.64), (0.36, 0.19) \rangle, \langle (0.72, 0.54), (0.28, 0.37) \rangle\}\}$

3. $\lambda(A, \hat{H}) = \{h_1, \{l_1, \langle(0.9744, 0.9375), (0.0081, 0.0081)\rangle,$
   $\langle(0.9919, 0.9744), (0.0016, 0.0081)\rangle l_2, \langle(0.9984, 0.9919), (0.0016, 0.0016)\rangle,$
   $\langle(0.9984, 0.9984), (0.0016, 0.0001)\rangle\}\}$
   $\{h_2 = \{l_1, \langle(0.9919, 0.9744), (0.0001, 0.0081)\rangle, \langle(0.9984, 0.9744), (0.0016, 0.0001)\rangle$
   $l_2, \langle(0.9984, 0.9984), (0.0016, 0.0001)\rangle, \langle(0.9999, 0.9999), (0.0001, 0.0001)\rangle\}\}\}$

4. $(A, \hat{H})^\lambda = \{h_1, \{l_1, \langle(0.1296, 0.0625), (0.7599, 0.7599)\rangle,$
   $\langle(0.2401, 0.1296), (0.5904, 0.7599)\rangle l_2, \langle(0.4096, 0.2401), (0.5904, 0.5904)\rangle,$
   $\langle(0.4096, 0.4096), (0.5904, 0.3439)\rangle\}\}$
   $\{h_2 = \{l_1, \langle(0.2401, 0.1296), (0.3439, 0.7599)\rangle, \langle(0.4096, 0.1296), (0.5904, 0.3439)\rangle$
   $l_2, \langle(0.4096, 0.4096), (0.5904, 0.3439)\rangle, \langle(0.6561, 0.6561), (0.3439, 0.3439)\rangle\}\}\}$

**Proposition 3.7.** *If $(A, \hat{H})$ and $(B, \hat{H})$ are two $\widehat{LOCLDFSS}$ then so $(A, \hat{H}) \oplus (B, \hat{H})$,*
*$(A, \hat{H}) \otimes (B, \hat{H})$, $\lambda(A, \hat{H})$ and $(A, \hat{H})^\lambda$*

*Proof.* Given $(A, \hat{H})$ and $(B, \hat{H})$ are $\widehat{LOCLDFSS}$, so we have
for $h_1 \leq h_2 \Rightarrow A(h_1) \leq A(h_2)$
also for $h_1 \leq h_2 \Rightarrow B(h_1) \leq B(h_2)$
To prove: $(D, \hat{H}) = (A, \hat{H}) \oplus (B, \hat{H})$ is also $\widehat{LOCLDFSS}$.
(*i.e.*), for any $h_1 \leq h_2$, we have to prove that $D(h_1) \leq D(h_2)$
since, $h_1 \leq h_2$ we have
$A(h_1) \leq A(h_2)$ and $B(h_1) \leq B(h_2)$
$\Rightarrow$

$\Gamma_{A(h_1)}(l_p) \leq \Gamma_{A(h_2)}(l_p) \qquad \Gamma_{B(h_1)}(l_p) \leq \Gamma_{B(h_2)}(l_p)$

$w_{\Gamma_{A(h_1)}}(l_p) \leq w_{\Gamma_{A(h_2)}}(l_p) \qquad w_{\Gamma_{B(h_1)}}(l_p) \leq w_{\Gamma_{B(h_2)}}(l_p)$

$\Delta_{A(h_1)}(l_p) \geq \Delta_{A(h_2)}(l_p) \qquad \Delta_{B(h_1)}(l_p) \geq \Delta_{B(h_2)}(l_p)$

$w_{\Delta_{A(h_1)}}(l_p) \geq w_{\Delta_{A(h_2)}}(l_p) \qquad w_{\Delta_{B(h_1)}}(l_p) \geq w_{\Delta_{B(h_2)}}(l_p)$

$\alpha^p_{A(h_1)} \leq \alpha^p_{A(h_2)} \qquad \alpha^p_{B(h_1)} \leq \alpha^p_{B(h_2)}$

$w_{\alpha^p_{A(h_1)}} \leq w_{\alpha^p_{A(h_2)}} \qquad w_{\alpha^p_{B(h_1)}} \leq w_{\alpha^p_{B(h_2)}}$

$\beta^p_{A(h_1)} \geq \beta^p_{A(h_2)} \qquad \beta^p_{B(h_1)} \geq \beta^p_{B(h_2)}$

$w_{\beta^p_{A(h_1)}} \geq w_{\beta^p_{A(h_2)}} \qquad \beta^p_{B(h_1)} \geq \beta^p_{B(h_2)}$

$\Rightarrow$

$\Gamma_{A(h_1)}(l_p) + \Gamma_{B(h_1)}(l_p) - \Gamma_{A(h_1)}(l_p)\Gamma_{B(h_1)}(l_p) \leq \Gamma_{A(h_2)}(l_p) + \Gamma_{B(h_2)}(l_p) - \Gamma_{A(h_2)}(l_p)\Gamma_{B(h_2)}(l_p)$

$w_{\Gamma_{A(h_1)}}(l_p) + w_{\Gamma_{B(h_1)}}(l_p) - w_{\Gamma_{A(h_1)}}(l_p)w_{\Gamma_{B(h_1)}}(l_p) \leq w_{\Gamma_{A(h_2)}}(l_p) + w_{\Gamma_{B(h_2)}}(l_p) - w_{\Gamma_{A(h_2)}}(l_p)w_{\Gamma_{B(h_2)}}(l_p)$

$\Delta_{A(h_1)}(l_p)\Delta_{B(h_1)}(l_p) \geq \Delta_{A(h_2)}(l_p)\Delta_{B(h_2)}(l_p)$

$w_{\Delta_{A(h_1)}}(l_p)w_{\Delta_{B(h_1)}}(l_p) \geq w_{\Delta_{A(h_2)}}(l_p)w_{\Delta_{B(h_2)}}(l_p)$

$\alpha^p_{A(h_1)} + \alpha^p_{B(h_1)} - \alpha^p_{A(h_1)}\alpha^p_{B(h_1)} \leq \alpha^p_{A(h_2)} + \alpha^p_{B(h_2)} - \alpha^p_{A(h_2)}\alpha^p_{B(h_2)}$

$w_{\alpha^p_{A(h_1)}} + w_{\alpha^p_{B(h_1)}} - w_{\alpha^p_{A(h_1)}}w_{\alpha^p_{B(h_1)}} \leq w_{\alpha^p_{A(h_2)}} + w_{\alpha^p_{B(h_2)}} - w_{\alpha^p_{A(h_2)}}w_{\alpha^p_{B(h_2)}}$

$\beta^p_{A(h_1)}\beta^p_{B(h_1)} \geq \beta^p_{A(h_2)}\beta^p_{B(h_2)}$

$$w_{\beta^p_{A(h_1)}} \, w_{\beta^p_{B(h_1)}} \geq w_{\beta^p_{A(h_2)}} \, w_{\beta^p_{B(h_2)}}$$

$$\Rightarrow D(h_1) \leq D(h_2) \text{ for } h_1 \leq h_2$$

Hence the proof.

The remaining can be proved in a similar manner.

**Theorem 3.8.** *1. The finite Algebraic sum of $LOC\widehat{LD}FSS$ is also a $LOC\widehat{LD}FSS$.*

*2. The finite Algebraic product of $LOC\widehat{LD}FSS$ is also a $LOC\widehat{LD}FSS$.*

*Proof.* The proof is evident.

**Proposition 3.9.** *Let us consider three $LOC\widehat{LD}FSS$ as $(A, \hat{H})$, $(B, \hat{H})$ and $(C, \hat{H})$. Then the following properties holds.*

*1. $(A, \hat{H}) \oplus (B, \hat{H}) = (B, \hat{H}) \oplus (A, \hat{H})$ (commutative under $\oplus$)*

*2. $(A, \hat{H}) \otimes (B, \hat{H}) = (B, \hat{H}) \otimes (A, \hat{H})$ (commutative under $\otimes$)*

*3. $((A, \hat{H}) \oplus (B, \hat{H})) \oplus (C, \hat{H}) = (A, \hat{H}) \oplus ((B, \hat{H}) \oplus (C, \hat{H}))$ (associative under $\oplus$)*

*4. $((A, \hat{H}) \otimes (B, \hat{H})) \otimes (C, \hat{H}) = (A, \hat{H}) \otimes ((B, \hat{H}) \otimes (C, \hat{H}))$ (associative under $\otimes$)*

*Proof.* The proof is evident.

# $LOC\widehat{LD}FSS$-DECISION MAKING PROCESS

This particular section includes the decision making process on $LOC\widehat{LD}FSS$. All through this section, assume that $\hat{L} = \{l_p : p = 1, 2, \ldots, n\}$ be the universal set, $\hat{H} = \{h_i : i = 1, 2, \ldots, m\}$ be the parameter set. A few of the requisite definitions are as follows.

## Score matrix and utility matrix for lattice ordered complex linear diophantine fuzzy soft matrix

**Definition 4.1.** *The Lattice Ordered Complex Linear Diophantine Fuzzy Soft Decision Matrix $(LOC\widehat{LD}FSM_{n \times m})$ is particularized by*

$$[M] = \left[ \left\langle \left( \Gamma^M_{h_{pi}}, w^M_{\Gamma_{h_{pi}}} \right), \left( \Delta^M_{h_{pi}}, w^M_{\Delta_{h_{pi}}} \right) \right\rangle, \left\langle \left( \alpha^M_{h_{pi}}, w^M_{\alpha_{h_{pi}}} \right), \left( \beta^M_{h_{pi}}, w^M_{\beta_{h_{pi}}} \right) \right\rangle \right]_{n \times m} = \begin{pmatrix} l_{11} & l_{12} & \cdots & l_{1m} \\ l_{21} & l_{22} & \cdots & l_{2m} \\ \vdots & \vdots & \ddots & \vdots \\ l_{n1} & l_{n2} & \cdots & l_{nm} \end{pmatrix}$$

*where $l_{pi} = \left( \langle (\Gamma_{h_i}(l_p), w_{\Gamma_{h_i}}(l_p)), (\Delta_{h_i}(l_p), w_{\Delta_{h_i}}(l_p)) \rangle, \langle (\alpha_{h_{pi}}, w_{\alpha_{h_{pi}}}), (\beta_{h_{pi}}, w_{\beta_{h_{pi}}}) \rangle \right)$, $p = 1, 2, \cdots, n$ and $i = 1, 2, \cdots, m$*

**Example 4.2.** *Consider a $LOC\widehat{LD}FSS$ $(A, \hat{H})$. Since it is a $LOC\widehat{LD}FSS$, for $h_1 \leq h_2 \Rightarrow A(h_1) \subseteq A(h_2)$.*

$$A(h_1) = \{l_1, \langle(0.5, 0.7), (0.3, 0.2)\rangle, \langle(0.6, 0.6), (0.4, 0.3)\rangle$$
$$l_2, \langle(0.7, 0.6), (0.3, 0.3)\rangle, \langle(0.8, 0.8), (0.2, 0.1)\rangle$$
$$l_3, \langle(0.6, 0.8), (0.3, 0.2)\rangle, \langle(0.8, 0.5), (0.2, 0.4)\rangle\}$$
$$A(h_2) = \{l_1, \langle(0.8, 0.8), (0.2, 0.1)\rangle, \langle(0.8, 0.7), (0.3, 0.2)\rangle$$
$$l_2, \langle(0.8, 0.7), (0.2, 0.2)\rangle, \langle(0.9, 0.9), (0.1, 0.1)\rangle$$
$$l_3, \langle(0.7, 0.9), (0.3, 0.1)\rangle, \langle(0.9, 0.6), (0.1, 0.1)\rangle\}$$

*The $\widehat{LOCLDFSM}_{n \times m}$ for above given $\widehat{LOCLDFSS}$ is as follows.*

$$M = \begin{bmatrix} [\langle(0.5, 0.7), (0.3, 0.2)\rangle, \langle(0.6, 0.6), (0.4, 0.3)\rangle] & [\langle(0.8, 0.8), (0.2, 0.1)\rangle, \langle(0.8, 0.7), (0.3, 0.2)\rangle] \\ [\langle(0.6, 0.8), (0.3, 0.2)\rangle, \langle(0.8, 0.5), (0.2, 0.4)\rangle] & [\langle(0.7, 0.6), (0.3, 0.3)\rangle, \langle(0.8, 0.8), (0.2, 0.1)\rangle] \\ [\langle(0.8, 0.7), (0.2, 0.2)\rangle, \langle(0.9, 0.9), (0.1, 0.1)\rangle] & [\langle(0.7, 0.9), (0.3, 0.1)\rangle, \langle(0.9, 0.6), (0.1, 0.1)\rangle] \end{bmatrix}$$

**Definition 4.3.** *The Score Matrix for $\widehat{LOCLDFSM}_{n \times m}$ is particularized as*

$$\hat{S}(M) = \frac{1}{4} \left( \left( \Gamma_{h_{pi}}^M - \Delta_{h_{pi}}^M \right) + \left( w_{\Gamma_{h_{pi}}}^M - w_{\Delta_{h_{pi}}}^M \right) + \left( \alpha_{h_{pi}}^M - \beta_{h_{pi}}^M \right) + \left( w_{\alpha_{h_{pi}}}^M - w_{\beta_{h_{pi}}}^M \right) \right),$$

$\forall\, p = 1, 2, \cdots, n$ *and* $i = 1, 2, \cdots, m$, *where $M$ is a $\widehat{LOCLDFSM}_{n \times m}$.*

**Example 4.4.** *The Score Matrix $\hat{S}(M)$ of Example 4.2 is*

$$\hat{S}(M) = \begin{bmatrix} 0.3 & 0.575 \\ 0.4 & 0.5 \\ 0.675 & 0.625 \end{bmatrix}$$

**Definition 4.5.** *The Utility Matrix is particularized as*

$$\hat{U}(M, I) = \hat{S}(M) - \hat{S}(I)$$

*where $M$ and $I$ are $\widehat{LOCLDFSM}_{n \times m}$.*

# MCDM TECHNIQUE BASED ON SCORE MATRIX AND UTILITY MATRIX

MCDM is one of the main decision-making techniques that helps in determining a suitable alternative by taking into account multiple criteria in a decision-making process. The intention of establishing criteria is to subsidize the decision-making process and the alternative selected based on the decision made should support the desired outcome. In this entry, the MCDM technique is incorporated on $\widehat{LOCLDFSS}$ to select a leading alternative amidst various aspirants in a decision in terms of ensuring the safety of the roads.

## Algorithm for the proposed technique to prefer an applicable car with adaptive cruise control system

The constructive steps for the proposed algorithm are illustrated in a schematic diagram and are symbolized in Fig. 1.

**Algorithm.**

Step 1: Input $LOC\widehat{L}DFSS$ and construct a $LOC\widehat{L}DFSM$.

Step 2: Estimate the Score Matrix $\hat{S}(M)$, $\hat{S}(I)$ and $\hat{S}(D)$ by using Definition 4.3.

Step 3: Determine the Utility Matrix $\hat{U}(M, I, D)$ by using Definition 4.5.

Step 4: Evaluate the Total Score Matrix.

Step 5: Rank the alternatives using Total Score Matrix.

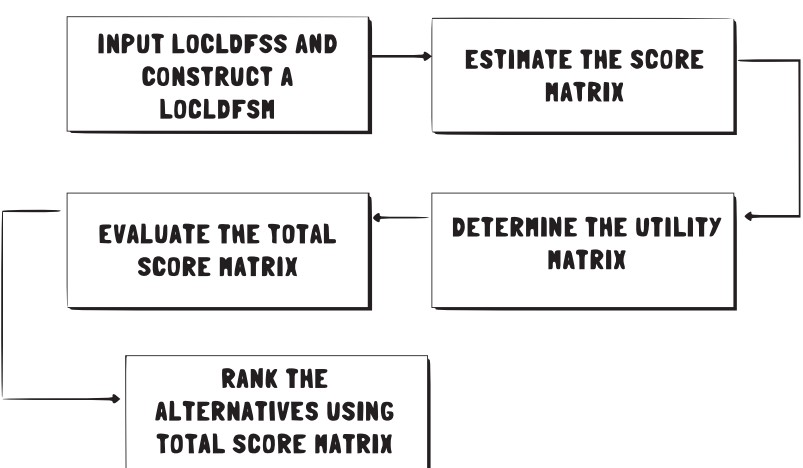

**Figure 1** **Constructive steps for the proposed algorithm.**

Safety is one of the most predominant aspects while purchasing a car and safety is closely associated with technology. For better road safety, cars in India have begun to incorporate ADAS technology into their systems. ADAS is an automated technology that uses sensors and cameras to detect nearby vehicles and respond accordingly to increase road safety. The importance of ADAS technology is described in Fig. 2.

In particular, ACC is a system in cars that uses ADAS technology. ACC accordingly adjusts the speed of a car as soon as it detects a slow-moving vehicle ahead to maintain a safe following distance. When the road ahead is clear, it automatically accelerates to the set input speed. ACC is ideal for highway roads.

ACC is a knowledgeable form of cruise control that automatically slows down and speeds up. Initially, the driver set the input speed. When the sensor detects any vehicle ahead, it instructs to stay behind at a particular distance by automatically slowing down. Once the vehicle ahead is apart from the range of a sensor it accordingly accelerates to the set-input speed.

Road accidents are frequently happening in India due to collisions. There is no guarantee for safer driving. ACC fills this gap and enables a convenient and safer driving experience by monitoring other vehicles and objects on the road with the use of sensors. It
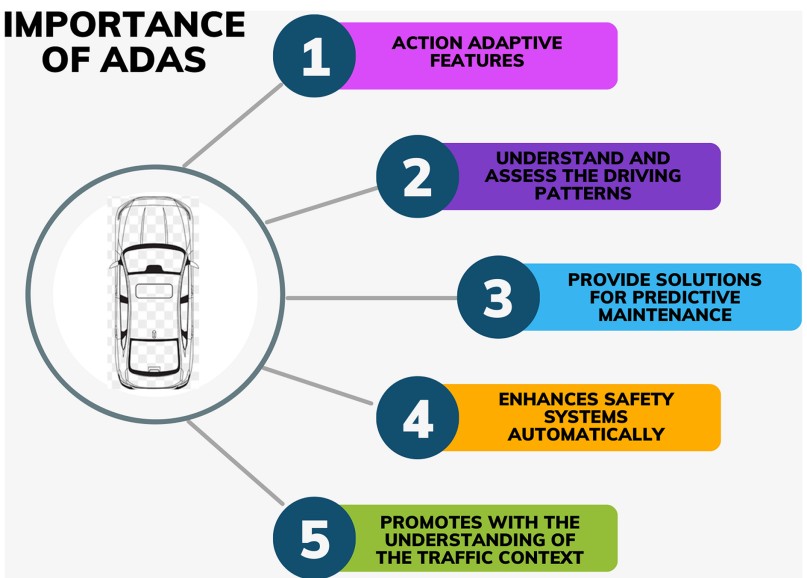

**Figure 2 Importance of ADAS technology.** The components in this figure were synthesized from the literature review and created by using Canva.

also helps the driver keep a steady vehicle speed. The main aim of this manuscript is to prefer an applicable car with ACC together with its latest model by $\widehat{LOCLDFSS}$ algorithm. Let $\{l_1, l_2, l_3, l_4\}$ symbolize the car with ACC. A balanced strategy to satisfy customer demands and industry requirements is ensured by giving careful consideration to cost for affordability, performance for efficiency, and sensor kinds for safety and technology integration. So let $\{h_1, h_2, h_3\}$ indicates the parameter set, where $h_1$ = Cost, $h_2$ = Performance and $h_3$ = Sensor type used. The fundamental aim of this manuscript is to prefer an applicable Car with ACC together with its latest model based on the expertise of the three experts namely $\{M, I, D\}$, where

1) M = Automotive Engineers: Technical specifications, structural design, and overall vehicle functionality are areas where engineers and designers are indispensable in the decision-making process.

2) I = Environmental Analysts: Emphasising on sustainability of the environment, evaluating the environmental effect of cars, and examining methods for implementing greener practices and technologies.

3) D = Safety and Compliance Analysts: Examining crash test outcomes, safety standards, and regulatory compliance to provide information on the level of security of a car.

The hierarchy of the parameters is $h_1 \leq h_2 \leq h_3$ and its diagrammatic representation is given in Fig. 3.

The classification of parameters is as follows.

1. The attribute "Cost" intimates that the alternative is "cheap" or "not cheap".

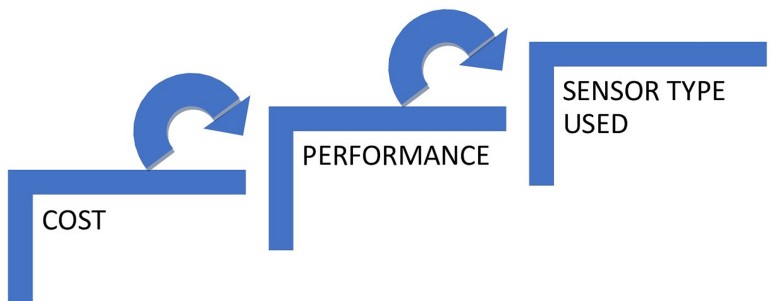

**Figure 3 The diagrammatic representation of hierarchy between the parameters.**

2. The attribute "Performance" intimates that the alternative is "good" or "bad".

3. The attribute "Sensor type used" intimates that the alternative is of "high standard" or "low standard".

The tabular representation is given below.

| Attributes | Characteristics of $\widehat{LOCLDFSS}$ |
|---|---|
| "Cost" | $(\langle \widehat{CVMC}, \widehat{CVNMC} \rangle, \langle cheap, not\ cheap \rangle)$ |
| "Performance" | $(\langle \widehat{CVMC}, \widehat{CVNMC} \rangle, \langle good, bad \rangle)$ |
| "Sensor type used" | $(\langle \widehat{CVMC}, \widehat{CVNMC} \rangle, \langle high, low \rangle)$ |

Expert D has determined that the alternative "Sensor type used" has a numerical value of $(\langle (0.8, 0.6), (0.2, 0.4) \rangle, \langle (0.6, 0.5), (0.4, 0.4) \rangle)$. This grade indicates that $l_1$ has a truth value of 0.8, a falsity value of 0.2, and a truth value of 0.6 together with its latest model, a falsity value of 0.4 together with its latest model for attribute "Sensor type used". The pair $\langle (0.6, 0.5), (0.4, 0.4) \rangle$ is regarded as a truth and falsity value for reference parameter, where we can apprise that $l_1$ should be of 0.6 value high standard, 0.4 value low standard, and 0.5 value high standard together with its latest model, 0.4 value low standard together with its latest model. All data are calculated similarly.

Now, an algorithm is presented for the wide range of determining a car with ACC. The $\widehat{LOCLDFSM}$ can be formulated from the following attributes value.

$$(A_M, \widehat{H}) = \begin{cases} \{A_M(h_1) = & \{(l_1, \langle (0.6, 0.7), (0.3, 0.2) \rangle, & (l_2, \langle (0.8, 0.7), (0.2, 0.2) \rangle, & (l_3, \langle (0.6, 0.9), (0.2, 0.1) \rangle, & (l_4, \langle (0.7, 0.6), (0.3, 0.3) \rangle, \\ & \langle (0.7, 0.7), (0.3, 0.3) \rangle) & \langle (0.7, 0.8), (0.3, 0.2) \rangle) & \langle (0.7, 0.8), (0.3, 0.1) \rangle) & \langle (0.7, 0.8), (0.3, 0.2) \rangle)\} \\ \{A_M(h_2) = & \{(l_1, \langle (0.7, 0.8), (0.2, 0.2) \rangle, & (l_2, \langle (0.8, 0.8), (0.1, 0.2) \rangle, & (l_3, \langle (0.8, 0.9), (0.1, 0.1) \rangle, & (l_4, \langle (0.8, 0.7), (0.1, 0.3) \rangle, \\ & \langle (0.7, 0.8), (0.3, 0.2) \rangle) & \langle (0.8, 0.8), (0.1, 0.2) \rangle) & \langle (0.8, 0.9), (0.2, 0.1) \rangle) & \langle (0.8, 0.8), (0.2, 0.2) \rangle)\} \\ \{A_M(h_3) = & \{(l_1, \langle (0.8, 0.9), (0.2, 0.1) \rangle, & (l_2, \langle (0.9, 0.9), (0.1, 0.1) \rangle, & (l_3, \langle (0.9, 0.9), (0.1, 0.1) \rangle, & (l_4, \langle (0.9, 0.8), (0.1, 0.2) \rangle, \\ & \langle (0.8, 0.8), (0.1, 0.1) \rangle) & \langle (0.9, 0.8), (0.1, 0.1) \rangle) & \langle (0.8, 0.9), (0.2, 0.1) \rangle) & \langle (0.8, 0.8), (0.2, 0.2) \rangle)\} \end{cases}$$

$$(A_I, \widehat{H}) = \begin{cases} \{A_I(h_1) = & \{(l_1, \langle(0.5,0.6),(0.4,0.3)\rangle, & (l_2, \langle(0.7,0.6),(0.3,0.3)\rangle, & (l_3, \langle(0.7,0.8),(0.3,0.2)\rangle, & (l_4, \langle(0.6,0.5),(0.4,0.4)\rangle, \\ & \langle(0.6,0.6),(0.4,0.4)\rangle) & \langle(0.6,0.7),(0.4,0.3)\rangle) & \langle(0.6,0.7),(0.4,0.2)\rangle) & \langle(0.6,0.8),(0.4,0.2)\rangle)\} \\ \{A_I(h_2) = & \{(l_1, \langle(0.6,0.7),(0.3,0.3)\rangle, & (l_2, \langle(0.8,0.7),(0.2,0.2)\rangle, & (l_3, \langle(0.8,0.8),(0.2,0.2)\rangle, & (l_4, \langle(0.7,0.6),(0.2,0.4)\rangle, \\ & \langle(0.6,0.7),(0.4,0.3)\rangle) & \langle(0.7,0.7),(0.2,0.3)\rangle) & \langle(0.7,0.8),(0.3,0.2)\rangle) & \langle(0.7,0.8),(0.3,0.2)\rangle)\} \\ \{A_I(h_3) = & \{(l_1, \langle(0.7,0.8),(0.3,0.2)\rangle, & (l_2, \langle(0.8,0.8),(0.2,0.1)\rangle, & (l_3, \langle(0.9,0.9),(0.1,0.1)\rangle, & (l_4, \langle(0.9,0.7),(0.1,0.3)\rangle, \\ & \langle(0.7,0.7),(0.2,0.2)\rangle) & \langle(0.9,0.8),(0.1,0.1)\rangle) & \langle(0.8,0.8),(0.2,0.2)\rangle) & \langle(0.8,0.9),(0.2,0.1)\rangle)\} \end{cases}$$

$$(A_D, \widehat{H}) = \begin{cases} \{A_D(h_1) = & \{(l_1, \langle(0.8,0.6),(0.2,0.4)\rangle, & (l_2, \langle(0.6,0.5),(0.2,0.4)\rangle, & (l_3, \langle(0.6,0.7),(0.4,0.3)\rangle, & (l_4, \langle(0.8,0.7),(0.2,0.3)\rangle, \\ & \langle(0.6,0.5),(0.4,0.4)\rangle) & \langle(0.6,0.6),(0.3,0.4)\rangle) & \langle(0.5,0.6),(0.4,0.3)\rangle) & \langle(0.5,0.7),(0.4,0.3)\rangle)\} \\ \{A_D(h_2) = & \{(l_1, \langle(0.8,0.6),(0.2,0.4)\rangle, & (l_2, \langle(0.7,0.6),(0.3,0.3)\rangle, & (l_3, \langle(0.7,0.7),(0.3,0.3)\rangle, & (l_4, \langle(0.8,0.7),(0.2,0.3)\rangle, \\ & \langle(0.6,0.6),(0.4,0.4)\rangle) & \langle(0.6,0.6),(0.3,0.4)\rangle) & \langle(0.6,0.7),(0.4,0.3)\rangle) & \langle(0.6,0.7),(0.4,0.3)\rangle)\} \\ \{A_D(h_3) = & \{(l_1, \langle(0.8,0.7),(0.2,0.3)\rangle, & (l_2, \langle(0.9,0.7),(0.1,0.3)\rangle, & (l_3, \langle(0.8,0.8),(0.2,0.2)\rangle, & (l_4, \langle(0.9,0.8),(0.1,0.2)\rangle, \\ & \langle(0.6,0.6),(0.3,0.3)\rangle) & \langle(0.8,0.7),(0.2,0.2)\rangle) & \langle(0.7,0.7),(0.3,0.3)\rangle) & \langle(0.7,0.8),(0.3,0.2)\rangle)\} \end{cases}$$

Step 1: The $LOC\widehat{LD}FSM$ is formulated from above mentioned $LOC\widehat{LD}FSS$.

$$M = \begin{bmatrix} [\langle(0.6,0.7),(0.3,0.2)\rangle, \langle(0.7,0.7),(0.3,0.3)\rangle] & [\langle(0.7,0.8),(0.2,0.2)\rangle, \langle(0.7,0.8),(0.3,0.2)\rangle] & [\langle(0.8,0.9),(0.2,0.1)\rangle, \langle(0.8,0.8),(0.1,0.1)\rangle] \\ [\langle(0.8,0.7),(0.2,0.2)\rangle, \langle(0.7,0.8),(0.3,0.2)\rangle] & [\langle(0.8,0.8),(0.1,0.2)\rangle, \langle(0.8,0.8),(0.1,0.2)\rangle] & [\langle(0.9,0.9),(0.1,0.1)\rangle, \langle(0.9,0.8),(0.1,0.1)\rangle] \\ [\langle(0.6,0.9),(0.2,0.1)\rangle, \langle(0.7,0.8),(0.3,0.1)\rangle] & [\langle(0.8,0.9),(0.1,0.1)\rangle, \langle(0.8,0.9),(0.2,0.1)\rangle] & [\langle(0.9,0.9),(0.1,0.1)\rangle, \langle(0.8,0.9),(0.2,0.1)\rangle] \\ [\langle(0.7,0.6),(0.3,0.3)\rangle, \langle(0.7,0.8),(0.3,0.2)\rangle] & [\langle(0.8,0.7),(0.1,0.3)\rangle, \langle(0.8,0.8),(0.2,0.2)\rangle] & [\langle(0.9,0.8),(0.1,0.2)\rangle, \langle(0.8,0.8),(0.2,0.2)\rangle] \end{bmatrix}$$

$$I = \begin{bmatrix} [\langle(0.5,0.6),(0.4,0.3)\rangle, \langle(0.6,0.6),(0.4,0.4)\rangle] & [\langle(0.6,0.7),(0.3,0.3)\rangle, \langle(0.6,0.7),(0.4,0.3)\rangle] & [\langle(0.7,0.8),(0.3,0.2)\rangle, \langle(0.7,0.7),(0.2,0.2)\rangle] \\ [\langle(0.7,0.6),(0.3,0.3)\rangle, \langle(0.6,0.7),(0.4,0.3)\rangle] & [\langle(0.8,0.7),(0.2,0.2)\rangle, \langle(0.7,0.7),(0.2,0.3)\rangle] & [\langle(0.8,0.8),(0.2,0.1)\rangle, \langle(0.9,0.8),(0.1,0.1)\rangle] \\ [\langle(0.7,0.8),(0.3,0.2)\rangle, \langle(0.6,0.7),(0.4,0.2)\rangle] & [\langle(0.8,0.8),(0.2,0.2)\rangle, \langle(0.7,0.8),(0.3,0.2)\rangle] & [\langle(0.9,0.9),(0.1,0.1)\rangle, \langle(0.8,0.8),(0.2,0.2)\rangle] \\ [\langle(0.6,0.5),(0.4,0.4)\rangle, \langle(0.6,0.8),(0.4,0.2)\rangle] & [\langle(0.7,0.6),(0.2,0.4)\rangle, \langle(0.7,0.8),(0.3,0.2)\rangle] & [\langle(0.9,0.7),(0.1,0.3)\rangle, \langle(0.8,0.9),(0.2,0.1)\rangle] \end{bmatrix}$$

$$D = \begin{bmatrix} [\langle(0.8,0.6),(0.2,0.4)\rangle, \langle(0.6,0.5),(0.4,0.4)\rangle] & [\langle(0.8,0.6),(0.2,0.4)\rangle, \langle(0.6,0.6),(0.4,0.4)\rangle] & [\langle(0.8,0.7),(0.2,0.3)\rangle, \langle(0.6,0.6),(0.3,0.3)\rangle] \\ [\langle(0.6,0.5),(0.3,0.4)\rangle, \langle(0.6,0.6),(0.3,0.4)\rangle] & [\langle(0.7,0.6),(0.3,0.3)\rangle, \langle(0.6,0.6),(0.3,0.4)\rangle] & [\langle(0.9,0.7),(0.1,0.3)\rangle, \langle(0.8,0.7),(0.2,0.2)\rangle] \\ [\langle(0.6,0.7),(0.4,0.3)\rangle, \langle(0.5,0.6),(0.4,0.3)\rangle] & [\langle(0.7,0.7),(0.3,0.3)\rangle, \langle(0.6,0.7),(0.4,0.3)\rangle] & [\langle(0.8,0.8),(0.2,0.2)\rangle, \langle(0.7,0.7),(0.3,0.3)\rangle] \\ [\langle(0.8,0.7),(0.2,0.3)\rangle, \langle(0.5,0.7),(0.4,0.3)\rangle] & [\langle(0.8,0.7),(0.2,0.3)\rangle, \langle(0.6,0.7),(0.4,0.3)\rangle] & [\langle(0.9,0.8),(0.1,0.2)\rangle, \langle(0.7,0.8),(0.3,0.2)\rangle] \end{bmatrix}$$

Step 2: Estimate the score matrix $\hat{S}(M)$, $\hat{S}(I)$ and $\hat{S}(D)$.

$$\hat{S}(M) = \begin{bmatrix} 0.4 & 0.525 & 0.7 \\ 0.525 & 0.65 & 0.775 \\ 0.575 & 0.725 & 0.75 \\ 0.425 & 0.575 & 0.65 \end{bmatrix}$$

$$\hat{S}(I) = \begin{bmatrix} 0.2 & 0.325 & 0.5 \\ 0.325 & 0.5 & 0.7 \\ 0.425 & 0.55 & 0.7 \\ 0.275 & 0.425 & 0.65 \end{bmatrix}$$

$$\hat{S}(D) = \begin{bmatrix} 0.275 & 0.3 & 0.4 \\ 0.225 & 0.3 & 0.575 \\ 0.25 & 0.35 & 0.5 \\ 0.375 & 0.4 & 0.6 \end{bmatrix}$$

Step 3: Determine the utility matrix as below.

$$\hat{U}(M, I, D) = \begin{bmatrix} 0.075 & 0.1 & 0.2 \\ 0.025 & 0.15 & 0.5 \\ 0.1 & 0.175 & 0.45 \\ 0.225 & 0.25 & 0.6 \end{bmatrix}$$

Step 4: Evaluation of the total score matrix.

$$Total\ Score\ Matrix = \begin{bmatrix} 0.375 \\ 0.675 \\ 0.725 \\ 1.075 \end{bmatrix}$$

Step 5: Ranking of the alternatives is $l_4 > l_3 > l_2 > l_1$

$l_4$ is the suitable car with ACC based on the expertise of the three experts by using $\widehat{LOCLDFSS}$-decision making process.

# COMPARATIVE ASSESSMENT WITH THE EXISTING METHODOLOGY

The proposed methodology is the most appropriate procedure to integrate vague and uncertain data in decision-making problems. To manifest the efficiency and adaptability of the proposed approach, a logical comparison between the proposed work with the various other existing works by using their algorithms was discussed to examine the viability of the current work.

## Comparative studies

In the method of *Borah, Neog & Sut (2012)* first, the parameter sets are converted into $\widehat{FSM}$. Then the cross product of $\widehat{FSM}$ should be evaluated. By the optimum matrix, the ranking of the alternatives was done to choose the suitable alternative. The above steps were conferred below.

### *Algorithm Borah, Neog & Sut (2012)*

Step 1: Decide on the parameter set.
Step 2: For every parameter set, establish $\widehat{FSM}$.
Step 3: Access the cross product of $\widehat{FSM}$.
Step 4: Compute the optimum subscript matrix.
Step 5: Quantify the suitable alternative having a maximum value.
   we now have

$$M \times I \times D = \begin{bmatrix} 0.24 + 0.336 + 0.448 \\ 0.336 + 0.448 + 0.648 \\ 0.252 + 0.448 + 0.648 \\ 0.336 + 0.448 + 0.729 \end{bmatrix}$$

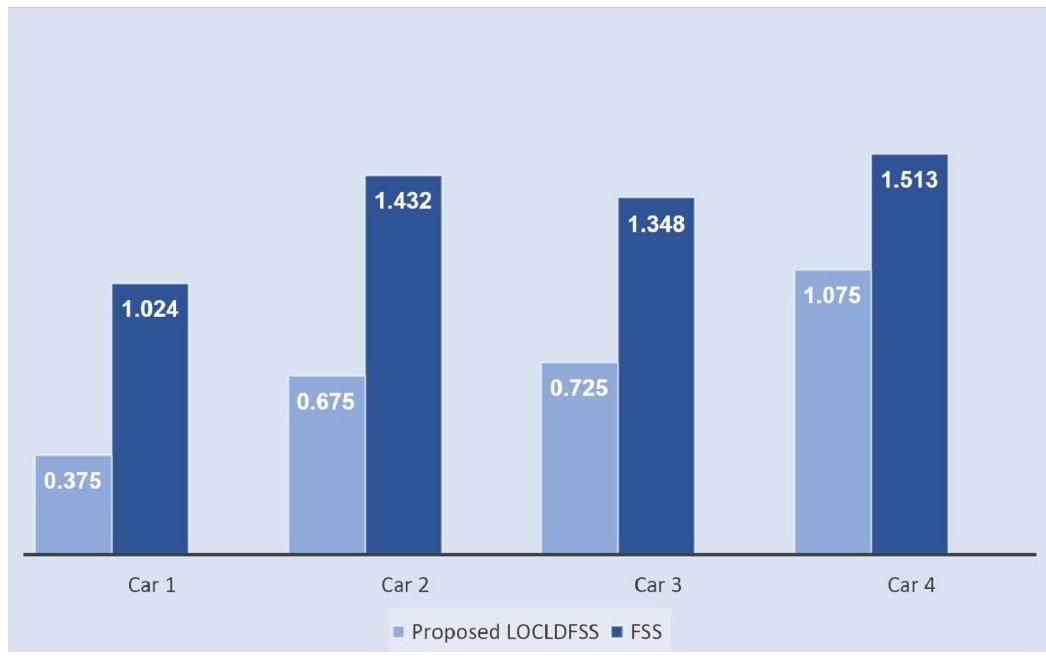

**Figure 4 The histogram of $\widehat{FSS}$ and proposed $\widehat{LOCLDFSS}$ algorithms.**

$$M \times I \times D = \begin{bmatrix} 1.024 \\ 1.432 \\ 1.348 \\ 1.513 \end{bmatrix}$$

Ordering of the alternatives is $l_4 > l_2 > l_3 > l_1$

The histogram of $\widehat{FSS}$ and proposed $\widehat{LOCLDFSS}$ algorithms is signified in Fig. 4. In the method of *Siddique et al. (2021)* first, construct the $\widehat{PFSM}$. Then the score and utility matrix were found. By the total score matrix, the ordering of an alternative was done to select a suitable alternative. The above steps are conferred below.

### Algorithm *Siddique et al. (2021)*

Step 1: Input $\widehat{PFSS}$ and construct $\widehat{PFSM}$.
Step 2: Determine the score matrix $\hat{S}(M)$, $\hat{S}(I)$, $\hat{S}(D)$.
Step 3: Detect the utility matrix.
Step 4: Find the total score matrix.
Step 5: By the total score matrix, rank the alternatives.

The score matrix is given below.

$$\hat{S}(M) = \begin{bmatrix} 0.27 & 0.45 & 0.6 \\ 0.6 & 0.63 & 0.8 \\ 0.32 & 0.63 & 0.8 \\ 0.4 & 0.63 & 0.8 \end{bmatrix}$$

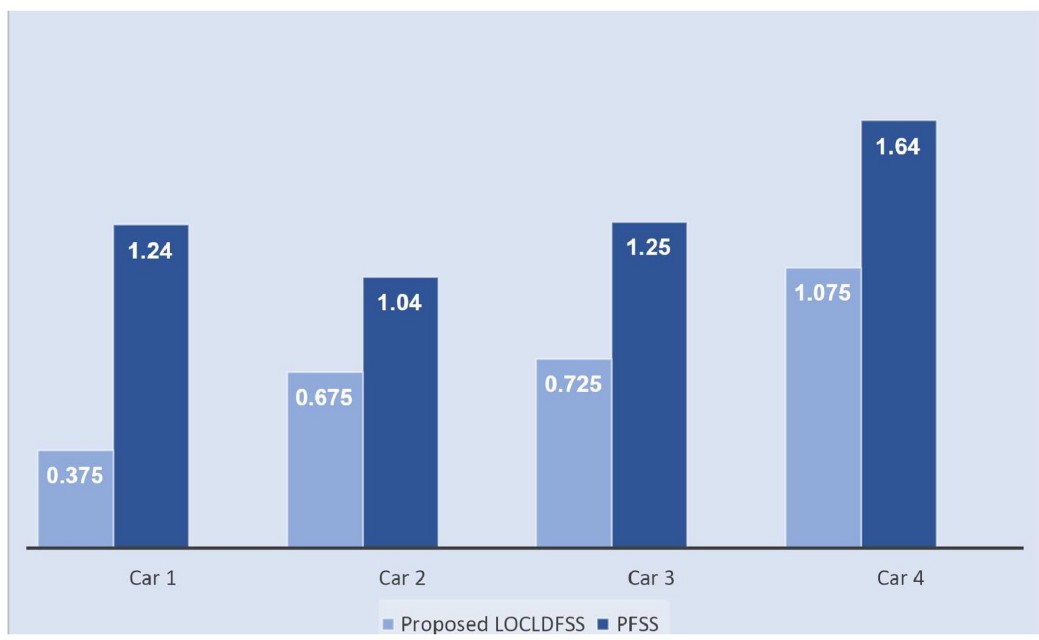

**Figure 5** The histogram of $\widehat{PFSS}$ and proposed $\widehat{LOCLDFSS}$ algorithms.

$$\hat{S}(I) = \begin{bmatrix} 0.09 & 0.27 & 0.4 \\ 0.4 & 0.6 & 0.6 \\ 0.4 & 0.6 & 0.8 \\ 0.2 & 0.45 & 0.8 \end{bmatrix}$$

$$\hat{S}(D) = \begin{bmatrix} 0.6 & 0.6 & 0.6 \\ 0.27 & 0.4 & 0.8 \\ 0.2 & 0.4 & 0.6 \\ 0.6 & 0.6 & 0.8 \end{bmatrix}$$

The utility matrix is as follows.

$$\hat{U}(M, I, D) = \begin{bmatrix} 0.42 & 0.42 & 0.4 \\ 0.07 & 0.37 & 0.6 \\ 0.28 & 0.37 & 0.6 \\ 0.4 & 0.42 & 0.8 \end{bmatrix}$$

$$Total\ score\ matrix = \begin{bmatrix} 1.24 \\ 1.04 \\ 1.25 \\ 1.62 \end{bmatrix}$$

Ordering of the alternatives is $l_4 > l_3 > l_1 > l_2$

The histogram of $\widehat{PFSS}$ and proposed $\widehat{LOCLDFSS}$ algorithms is illustrated in Fig. 5.

The proposed method can also be compared with the existing algorithm which has its input set as $\widehat{IFSS}$, if the sum of $\widehat{MC}$ and $\widehat{NMC}$ lie in [0,1]. The comparison table for the

**Table 1 Comparison table.**

Comparison table

| Models | Ranking of the alternatives | Suitable car with ACC |
|---|---|---|
| Proposed $LOC\widehat{LD}FSS$ | $l_4 > l_3 > l_2 > l_1$ | $l_4$ |
| Borah, Neog & Sut (2012) | $l_4 > l_2 > l_3 > l_1$ | $l_4$ |
| Siddique et al. (2021) | $l_4 > l_3 > l_1 > l_2$ | $l_4$ |

algorithms in "Algorithm for the Proposed Technique to Prefer an Applicable Car with Adaptive Cruise Control system", "Algorithm *Borah, Neog & Sut (2012)*", and "Algorithm *Siddique et al. (2021)*" are given in Table 1

In comparing our suggested $LOC\widehat{LD}FSS$ with the algorithms of *Borah, Neog & Sut (2012)* and *Siddique et al. (2021)* in determining a car equipped with Adaptive Cruise Control (ACC), several significant features become apparent:

**Accuracy:** In identifying a suitable car with ACC, our $LOC\widehat{LD}FSS$ method performs comparable accuracy to algorithms developed by *Borah, Neog & Sut (2012)* and *Siddique et al. (2021)*. This suggests that all three approaches may successfully capture the intricacies of the standards used to make decisions and offer precise suggestions.

**Computational efficiency:** Our suggested $LOC\widehat{LD}FSS$ makes sure it guarantees prompt decision-making without sacrificing accuracy by processing large, complicated data sets efficiently.

**Generalization and scalability:** The proposed $LOC\widehat{LD}FSS$ exhibits flexibility in a range of datasets and decision-making scenarios, guaranteeing strong recommendations that go beyond ACC selection. Furthermore, when data complexity increases, its scalability guarantees continued performance by facilitating the effective processing of bigger datasets and complicated decision scenarios. Overall, $LOC\widehat{LD}FSS$ seems to be a flexible and trustworthy framework for making decisions, well-suited to handle a broad range of real-world applications beyond the selection of an automobile.

**Ease of interpretation:** Our $LOC\widehat{LD}FSS$ algorithm makes decision-making accessible to decision-makers of different skill levels by providing a simple and easy-to-understand framework for evaluating decision outcomes. $LOC\widehat{LD}FSS$ enables well-informed decision-making by offering transparent insights into the decision-making process.

In summary, our $LOC\widehat{LD}FSS$ algorithm has significant advantages over the algorithms of *Borah, Neog & Sut (2012)* and *Siddique et al. (2021)* despite achieving similar optimal selections. These advantages include balanced accuracy, effective data processing, flexibility, scalability, and ease of understanding. Due to these characteristics, $LOC\widehat{LD}FSS$ is positioned as a strong and adaptable framework for making decisions when choosing automobiles with ACC and handling other difficult situations.

## Advantages of the proposed methodology

As demonstrated by the comparison analysis mentioned above, the suggested approach to solving the decision-making issues provides the following advantages concerning the existing ones.

1) The classical $\widehat{FS}$ and $\widehat{PFS}$ are all particular instances of the $\widehat{LOCLDFSS}$, as was previously mentioned. Because of the limitations of the current theories, most problems in daily life cannot be solved when an expert states their preferences for the elements. One generalized theory that can deal with partial, ambiguous, and inconsistent information that is frequently present in real-world scenarios is $\widehat{LOCLDFSS}$. As a result, current studies are better suited to address engineering design and real-world issues than previous research.

2) It has been recognized that the proposed utility and score matrices in the $\widehat{LOCLDFSS}$ environment contribute to the body of knowledge already in existence and aid in the modeling of some real-world scenarios that the existing literature is unable to manage. However, the suggested approach is a better way to address issues because it may address the shortcomings of the current approaches.

## Limitations and practical implication

Data dependency: The algorithms' effectiveness is highly dependent on the quality and quantity of available data, ultimately resulting in inappropriate recommendations if data is inadequate or biased.

Computational complexity: The intricate nature of $\widehat{LOCLDFSS}$ and related algorithms may be difficult to develop and may require a large amount of computer power and knowledge. This could make them less useful in real-world scenarios where resources are limited.

## CONCLUSION

The MCDM technique is one of the classical methods which precisely capture the uncertainties. A team of experts in the decision-making process enlarges the credibility of outcomes. In this manuscript, the concept of $\widehat{LOCLDFSS}$ is initiated where the order exists between the elements of the parameter set. The space of this concept is immense due to the presence of amplitude term and phase term. We have illustrated a few operations with supportive examples and theorems. An algorithm has been developed using the MCDM technique incorporating $\widehat{LOCLDFSS}$. The proposed algorithm is efficient because the algorithm rule is made in accordance with the Score function.

To validate the theory, upon reaching the application phase, we have the ACC system which helps to reduce road safety issues like the possibility of crashes caused by different driving speeds. ACC is a technological solution that improves safety by minimizing the effects of human factors like delayed reactions or erratic driving habits. It does this by using sophisticated sensors and algorithms to maintain a safe following distance and automatically adjusting the vehicle's speed in response to traffic conditions. Because ACC facilitates better traffic flow and reduces abrupt stops, it lessens the possibility of rear-end crashes, improving overall road safety. Having considered this problem, $\widehat{LOCLDFSS}$ in the MCDM technique has been utilized to identify an application car with ACC together with its latest model.

In comparison to the existing algorithms, the proposed findings can be considerably more stable and feasible. The proposed findings can be adapted to address $\widehat{MCDM}$ problems in various uncertain environments. This developed method can also be profitably applied in the fields of agriculture and medicine.

The future directions for research that are indicated, such as the analysis of operators, the application of the VIKOR method, the inspection of distance metrics, and the incorporation of entropy measures into the $\widehat{LOCLDFSS}$ framework, outline a comprehensive strategy to improve the proposed methodology $\widehat{LOCLDFSS}s$ performance and robustness. By exploring these facets, it is anticipated that $\widehat{LOCLDFSS}$ would become more flexible in a variety of decision-making situations, leading to more sophisticated and knowledgeable choice results. The goal of this investigation is to improve the methodology's handling of uncertain and complex data structures, which will ultimately increase its applicability in a variety of contexts.

### Funding
The article has been written with the joint financial support of RUSA-Phase 2.0 grant sanctioned vide letter No.F 24-51/2014-U, Policy (TN Multi-Gen), Dept. of Edn. Govt. of India, Dt. 09.10.2018, DST-PURSE 2nd Phase programme vide letter No. SR/PURSE Phase 2/38 (G) Dt. 21.02.2017 and DST (FIST–level I) 657876570 vide letter No.SR/FIST/MS-I/2018/17Dt. The funders had no role in study design, data collection and analysis, decision to publish, or preparation of the manuscript.

### Grant Disclosures
The following grant information was disclosed by the authors:
RUSA-Phase 2.0 grant sanctioned vide letter No.F 24-51/2014-U, Policy (TN Multi-Gen), Dept. of Edn. Govt. of India, Dt. 09.10.2018, DST-PURSE.
2nd Phase programme vide letter No. SR/PURSE Phase 2/38 (G) Dt. 21.02.2017 and DST (FIST–level I) 657876570 vide letter No.SR/FIST/MS-I/2018/17Dt.

### Competing Interests
Željko Stević is an Academic Editor for PeerJ Computer Science.

### Author Contributions
- K. Ashma Banu conceived and designed the experiments, performed the experiments, analyzed the data, performed the computation work, prepared figures and/or tables, authored or reviewed drafts of the article, and approved the final draft.
- J. Vimala conceived and designed the experiments, performed the experiments, analyzed the data, performed the computation work, prepared figures and/or tables, authored or reviewed drafts of the article, and approved the final draft.
- Nasreen Kausar conceived and designed the experiments, performed the experiments, analyzed the data, performed the computation work, authored or reviewed drafts of the article, and approved the final draft.

- Željko Stević performed the experiments, analyzed the data, performed the computation work, authored or reviewed drafts of the article, and approved the final draft.

## Data Availability

The raw data are available in the Supplemental Files.

## Supplemental Information

Supplemental information for this article can be found online at http://dx.doi.org/10.7717/peerj-cs.2165#supplemental-information.

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
