# Peer review of "Optimizing road safety: integrated analysis of motorized vehicle using lattice ordered complex linear diophantine fuzzy soft set"

_PeerJ Computer Science, doi:10.7717/peerj-cs.2165_

## Round 0.1 · original submission · Major Revisions

In the opinions of reviewers and mine, this paper should undergo a major revision.

**Language Note:** The review process has identified that the English language must be improved. PeerJ can provide language editing services - please contact us at [email protected] for pricing (be sure to provide your manuscript number and title). Alternatively, you should make your own arrangements to improve the language quality and provide details in your response letter. – PeerJ Staff

Reviewer 1 ·

Basic reporting

check comments

Experimental design

check comments

Validity of the findings

check comments

Additional comments

The paper "A Systematic Analysis of Motorized Vehicles Using Lattice Ordered Complex Linear Diophantine Fuzzy Soft Set" explores the integration of Lattice Ordered Complex Linear Diophantine Fuzzy Soft Sets (LOCLDFSS) with Multi-Criteria Decision Making (MCDM) techniques to enhance decision-making processes related to Adaptive Cruise Control (ACC) systems in vehicles. By utilizing complex mathematical structures to represent and analyze the fuzziness and uncertainties inherent in the parameters influencing ACC systems, the study aims to provide a more refined and accurate framework for selecting optimal vehicle configurations. The authors propose novel mathematical operations and an algorithm to facilitate this decision-making process, showcasing the potential of LOCLDFSS in improving road safety and vehicle performance. However, the paper acknowledges the need for further refinement and validation of its proposed methodologies.
• Mathematical Rigor and Clarity: The paper introduces complex mathematical concepts and operations involving LOCLDFSS. However, the explanations and justifications for these mathematical structures are somewhat brief. Could the authors provide a more detailed theoretical foundation for the choice of LOCLDFSS and its superiority or advantages over other fuzzy set theories in this context?
• Experimental Validation: The application of the proposed methodology to select optimal vehicle configurations with ACC is innovative. However, the empirical validation of the proposed model seems limited to theoretical examples. Are there real-world data or case studies that could be used to demonstrate the practical applicability and effectiveness of the proposed system in real-life ACC selection scenarios?
• Comparison with Existing Methodologies: While the paper briefly mentions the novelty of its approach compared to existing methods, a detailed comparative analysis would strengthen its contributions. Specifically, how does the proposed LOCLDFSS-based MCDM technique perform relative to traditional fuzzy set and decision-making approaches in terms of accuracy, computational efficiency, and ease of interpretation?
• Sensitivity Analysis: Given the complex interplay of parameters in ACC systems and their representation within a fuzzy set framework, the sensitivity of the decision-making process to variations in these parameters is a critical aspect. Can the authors discuss how changes in input parameters or weights affect the outcomes of the proposed model? This analysis could highlight the robustness and reliability of the model under different scenarios.
• Generalization and Scalability: The focus on ACC systems in vehicles is highly relevant, but the potential of the proposed methodology extends beyond this application. Can the authors discuss the generalizability of their approach to other decision-making problems within and outside the automotive domain? Additionally, considerations regarding the scalability of the proposed model, especially in dealing with larger datasets or more complex decision-making scenarios, would be valuable.
• User Interface and Implementation Challenges: Implementing sophisticated mathematical models in practical decision-making tools requires careful consideration of user interfaces and the computational infrastructure needed. Can the authors comment on the potential challenges and solutions for integrating their model into real-world decision-support systems?

Reviewer 2 ·

Basic reporting

- The title of the paper can be modified with an aim that more deeply describes the content of the paper.

- Modify the abstract with an emphasis on main results, and contributions.

- The literature review presented through the introduction should be enriched with more relevant studies. Include references from methodological and professional aspects too.

Experimental design

no comment

Validity of the findings

- Table 1 needs more explanations. Please describe all the advantages of your proposed approach in comparison to others used in your paper.

- Write the limitations of your model and practical implications.

- Guidelines for future researchers are not well presented, please include more guidelines.

Additional comments

- Correct typos errors... The development of ADAS technology has been the focus of numerous researchers(5; 13; 15; 16; 22). Thereby, Zadeh(35) advanced the...Attanasov(2)... etc.

- Citations style is rare, a combination of common brackets and names in certain places is not correct. Please read the instructions and correct it. Seems that should be APA style of citations.

- Figure 3 should be modified. Please write all stages of your research and present it on diagram research flow.

Reviewer 3 ·

Basic reporting

The paper has been revised. The authors have tried to reply to each objective comment provided by reviewers. Also, I can see that WS correlation coefficient has been added in the revision, which additionally improved the paper.

Experimental design

It was improved. No new comentsts.

Validity of the findings

It was improved. No new comentsts.

Additional comments

Also, some comments have not been adopted provided by reviewer 1. The authors well discussed why they did not adopt them. In general, the paper has a good structure with the proposed novel rough MCDM method.

The paper can be considered for publication due to a lot of advantages. However, my contributions in reviewing this paper should be manifested through the next suggestions:
- Give more focus on motivation at the beginning of the introduction.
- If you add a Table in the Literature review you can improve this part and provide more readable content for readers.
- White words in the diagram of research (Figure 1) should be replaced with black.
- In the fourth section you can give some calculation examples, especially for the novel proposed rough WISP method.
- Results must be more discussed.
- Some recent references can be added.

---

## Round 0.2 · accepted · Accept

In the opinions of original reviewers, this revised paper can be accepted.

Reviewer 2 ·

Basic reporting

Authors addressed all issues identified in the previous review round

Experimental design

Authors addressed all issues identified in the previous review round

Validity of the findings

Authors addressed all issues identified in the previous review round

Additional comments

Considering the improvements of the manuscript quality, I congratulate the authors for the excellent work and I recommend the paper for publication.

Reviewer 3 ·

Basic reporting

no comment

Experimental design

no comment

Validity of the findings

no comment

Additional comments

The paper has been improved. Therefore, I recommend to accept it in its current form.